# Between Class and Career: Applying the Job Demands–Resources Model to Working College Students

**DOI:** 10.3390/bs16010061

**Published:** 2025-12-30

**Authors:** Kristen M. Tooley, Stephanie L. Dailey, Evan Schmiedehaus, Millie Cordaro, Natalie Dwyer, Dacey Jerkins, Krista Howard

**Affiliations:** 1Department of Psychology, Texas State University, San Marcos, TX 78666, USA; mc71@txstate.edu (M.C.); krista.howard@txstate.edu (K.H.); 2Department of Communication Studies, Texas State University, San Marcos, TX 78666, USA; dailey@txstate.edu; 3Department of Philosophy, Texas State University, San Marcos, TX 78666, USA; jes330@txstate.edu

**Keywords:** engagement, burnout, attrition, working college students, Job Demands–Resources model

## Abstract

The current study assessed organizational and psychosocial factors related to intentions to quit in American working college undergraduates (*N* = 382; mean age = 19 years; ~80% female). Students were surveyed on organizational scales (e.g., organizational identification, perceived support, work–life conflict, and intentions to quit) and psychosocial scales (e.g., perceived stress, social support, burnout, and mental health conditions). Variables significantly correlated with intent to quit at the bivariate level were included in an exploratory multiple regression model. The results indicated that burnout, engagement, organizational identification, perceived social support, and life–work conflict were uniquely predictive of intention to quit. A subsequent path analysis based on the Job Demands–Resources model revealed a good fit to the student data: demands (i.e., work–life conflict, perceived stress) and resources (organizational support and identification) predicted burnout and engagement, which in turn predicted intent to quit (along with a direct path from organizational support). This model can therefore explain behavior in both traditional and college undergraduate employees. In order to retain these employees, organizations should invest in practices that increase organizational identification and perceived support, as well as initiatives that help students mitigate the increased risks of stress and burnout associated with working while in college.

## 1. Introduction

College offers students opportunities to gain new knowledge and develop as individuals, while also creating significant stressors, such as adjusting to a new living situation, social support system ([15]), and financial strain. The rising cost of a college education ([7]) has resulted in many college students seeking simultaneous employment ([55]), often in jobs that offer few benefits and have undesirable working conditions. These are factors known to contribute to intentions to quit and turnover in the general workforce ([72]), yet complementary research on college student employees is currently lacking. The current study seeks to better understand working undergraduates’ intentions to quit, exploring both the role of employer organizational factors and employee psychosocial factors in job turnover.

Employment during college is appealing because it brings advantages like helping students pay for tuition and related expenses. Graduates who work during college also tend to earn higher salaries, even after controlling for other variables ([55]). As [58] ([58]) explain, work can help students apply what they learn in class as well as develop soft skills and networks that are useful in future professional endeavors. Several studies have also demonstrated that student employment helps with persistence towards degree completion ([37]; [45]). Research on the effects of employment on GPA and academic performance is mixed, but evidence suggests that part-time employment and working on campus yield greater benefits than full-time work off-campus ([20]; [48]). This nuance is important to consider, given that most students in countries that do not offer national tuition assistance work more than part-time ([55]), whereas students in many European countries devote fewer than 20 h per week working outside school ([25]). The current study focuses on a sample of college undergraduates from the United States, where the vast majority work for financial reasons (Perna).

Previous college employment studies have largely examined academic performance and retention issues ([35]), where working long hours and perceived financial burden are associated with decreases in these outcomes ([2]; [56]). Yet, a better understanding of the occupational behaviors of this unique population is needed if students are to reap the benefits of employment while navigating the challenges of working while in college. For example, the psychological contract, the reciprocal agreement between workers and their employer ([62]), differs between traditional full-time employees and college workers, who often view their temporary, unstable, and lower-wage employment during college as less than a “real job” ([12]). Additionally, they will have to navigate increasing employee disengagement ([1]; [26]), which recent polls suggest is especially pronounced among Generation Z employees ([32]).

Our approach to this gap in the literature is grounded in the Job Demands–Resources model ([18]), a theoretical framework that proposes employees experience negative (i.e., demands) and positive (i.e., resources) physical, organizational, psychological, or social aspects of work that affect employee outcomes. More job demands and fewer resources lead to employee burnout and disengagement. Higher resources foster motivation and productivity ([63]). Past scholarship has applied this model to a wide range of occupations to improve traditional, full-time employees’ well-being and performance ([5]). We apply this model to working college undergraduates, exploring both job resources (e.g., organizational identification and support) and demands (e.g., strain from managing work and personal obligations). We also apply this model to students in their academic environments, exploring whether resources like social support and the demands of being a student (i.e., perceived stress) relate to their employment decisions about quitting.

Employees’ organizational identification, or their perception of oneness or belongingness with their organization ([34]), is an important resource that has been found to decrease turnover intentions in traditional employees ([76]). Organizational identification can also mediate the relationship between employees’ perceptions of organizational practices and counterproductive working behavior ([17]), as employees who identify strongly with their employers respond better to undesirable workplace decisions such as necessary overtime. On the flip side, workplace ambivalence is associated with physical and psychological strain ([27]), which can lead to feeling disconnected from work and deciding to quit ([36]). Prior research suggests that identification processes differ in non-traditional work settings, such as internships ([16]), but little is known about organizational identification in college student workers. Thus, examining organizational identification in this novel population may help improve retention strategies aimed at undergraduate employees.

Importantly, college students face unique job demands, where they may struggle to balance the demands of employment and college coursework. This can manifest as difficulties completing work tasks due to life demands and/or difficulty fulfilling life responsibilities due to work demands. Managing these competing priorities can lead to low motivation, increased absenteeism, social loafing, and depressive symptoms ([70]). While most research on work–life balance has focused on traditional employees ([51]), the principles likely apply to working college students. A recent study by [66] ([66]) found that better work–life balance is associated with higher motivation, increased productivity, and reduced absenteeism. Similarly, [73] ([73]) demonstrated that individuals managing dual roles exhibited more positive behaviors when work–life balance was achieved. Understanding how work–life conflict impacts undergraduate employees is therefore crucial for predicting their intentions to quit.

College undergraduates also report significant levels of stress ([53]), which is a known predictor of health and well-being ([8]). Juggling academic responsibilities, financial concerns, and family pressures ([6]; [22]; [31]) can decrease quality of life ([39]). According to the American College Health Association—National College Health Assessment ([3]), 76.4% of college students reported moderate to high levels of stress within the last 30 days. Rates of anxiety, a serious health issue among college students, continue to increase since the COVID-19 pandemic ([9]; [64]; [75]). Importantly, stress experienced during the college years is predictive of mental health disorders ([31]), with the onset of depression and anxiety frequently presenting during this period ([40]). Moreover, inadequate time for social support that mitigates stress can lead to increased depressive symptoms ([50]; [70]).

College undergraduates are increasingly choosing to work, which can increase stressors and psychological vulnerability to mental health symptoms and disorders ([6]; [38]). Employed college students are also an overlooked yet particularly vulnerable population to burnout, given the “working” time required for both coursework and employment. Burnout is a multi-dimensional experience including emotional exhaustion, depersonalization, and reduced personal achievement, and is a key predictor of intentions to quit ([44]; [47]). While higher subjective workload is associated with higher levels of burnout ([74]), extracurricular activities ([78]) and social support ([79]) have been associated with decreased levels of burnout. For working college students, it is therefore important to consider mental health disorders alongside other variables when trying to identify which demands/resources are most related to intentions to quit.

Stress among working college students can be compounded when they lack adequate workplace social support ([77]), such as emotional, instrumental, or informational support ([29]). Workplace-supported employees are more likely to feel connected to their employer, have positive employee sentiment, and reduced stress, which can reduce burnout and work–life imbalance ([33]; [79]). These insights may be especially vital for student employees, who likely rely on multiple sources of social support while managing their various responsibilities.

In sum, the present study aims to comprehensively explore the occupational health of employed college undergraduates, considering employee organizational factors and psychosocial factors to identify those that can be leveraged to improve occupational and psychological health. Guided by the Job Demands–Resources model, we explore several factors potentially related to intentions to quit, including presenteeism, vacation shaming, work–life and life–work conflict, perceived stress, burnout (as demands variables), and organizational identification, organizational support, social support, work engagement and self-esteem (as resource variables). While mental health disorders are neither resources nor demands, we also consider Generalized Anxiety Disorder and Major Depressive Disorder measures in order to control for any effects of poor mental health on predicted outcomes of intentions to quit. Specifically, we use multiple regression and path analysis to test the following predictions about our working undergraduate sample:Increased levels of stress, work–life conflict, absenteeism, and burnout will be associated with increased intentions to quit.Increased social support, self-esteem, engagement, and organizational identification will be associated with decreased intentions to quit.Demands (e.g., stress and work–life conflict) will be positively associated with burnout, which in turn will be positively associated with intention to quit.Resources (e.g., organizational support and identification) will be positively associated with engagement, which in turn will be negatively associated with intention to quit.

## 2. Materials and Methods

### 2.1. Participants and Procedures

Participants (*N* = 382) were comprised of a sub-sample of self-identified working college undergraduates from a larger study on the occupational health of working adults that focused on psychological and occupational factors related to engagement/disengagement, intention to quit, and perceptions of quiet quitting. This study was approved by the Texas State University (San Marcos, TX, USA) Institutional Review Board (IRB) (Approval # 8864). All participants gave informed consent by clicking an option on the consent form page before responding to any survey questions, and were compensated with course credit.

Students enrolled in Introductory Psychology during the 2023 Fall semester were required to engage in research experience options, one of which was to participate in a research study. Students who met the study’s criteria (being employed while being enrolled in college courses) and chose to participate were asked to complete an online, anonymous survey in Qualtrics. The gender make-up of the sample was: 18.3% (69) males; 80.1% (302) females; 1.1% (4) non-binary/third gender; and 0.5% (2) not disclosed. The racial/ethnic demographics include 10.3% (39) Black; 38.7% (146) White; 41.4% (156) Hispanic/Latino/a/x; and 9.5% (36) Mixed or Additional Races. The mean age of this sample was 19.0 years (SD = 2.1).

### 2.2. Measures

#### 2.2.1. Demographics

Age, gender identity, race, and ethnicity were surveyed. Work demographics included years at current job, hours/week working, number of days working per week, and base salary.

#### 2.2.2. Occupational Factors

Organizational Identification was measured with the shortened version of the Organizational Identification Questionnaire (OIQ; [11]) and was used to assess participants’ attachment to their employer organization. This scale consisted of 4 items on a 5-point Likert scale, ranging from strongly disagree to strongly agree. An example item included: “I feel I have a lot in common with others in this organization.” The internal consistency of the OIQ is good (Cronbach’s alpha = 0.746; M = 13.6, SD = 3.2).

Three additional measures of organizational identification (Ambivalent Identification, Disidentification, and Neutral Identification) were included as research suggests identification can take on different dimensions ([34]). Ambivalent Identification was used for assessing participants’ ambivalence towards their employer organization. This scale consisted of 6 items on a 5-point Likert scale, ranging from strongly disagree to strongly agree. An example item included: “I have mixed feelings about my affiliation with this organization.” Disidentification was used for assessing participants’ opposition towards their employer organization. This scale consisted of 6 items on a 5-point Likert scale, ranging from strongly disagree to strongly agree. An example item included: “I am embarrassed to be part of this organization.” Neutral Identification was used for assessing tensions in participants’ attachment to the employer organization. This scale consisted of 6 items on a 5-point Likert scale, ranging from strongly disagree to strongly agree. An example item included: “It really doesn’t matter to me what happens to this organization.” The internal consistencies of each of the organizational subscales are good (Ambivalent Identification Cronbach’s alpha = 0.911; M = 12.2, SD = 34.9; Disidentification Cronbach’s alpha = 0.917; M = 9.3, SD = 4.7; and Neutral Identification Cronbach’s alpha = 0.929; M = 15.0, SD = 6.1).

The Survey of Perceived Organizational Support—Shortened Version (SPOS) was used for assessing the degree to which the student employee perceives that their employer organization shows concern for their well-being ([21]; [59]). This scale consisted of 8 items on a 5-point Likert scale, ranging from strongly disagree to strongly agree. An example item included: “My organization really cares about my well-being.” The internal consistency of the SPOS is good (Cronbach’s alpha = 0.891; M = 28.8, SD = 7.2).

Work Engagement was assessed using the Intellectual, Social, and Affective Engagement Scale (ISA; 9-items; [67]). This assessment has three subscales: Intellectual Engagement, Social Engagement, and Affective Engagement, measured on a 7-point Likert scale from strongly disagree to strongly agree. The internal consistency of the ISA is good (Cronbach’s alpha = 0.886; M = 48.1, SD = 8.6).

The Work–Family Conflict Scale (9-items; [51]) assesses both work–life conflict (WFC; the degree to which work interferes with the participants’ life) and family–work conflict (FWC; the degree to which family interferes with the participants’ work), which are related but separable constructs. Each item is on a 7-point Likert scale with responses ranging from strongly disagree to strongly agree. An example item from the WFC is, “The demands of my work interfere with my personal life,” and an example item from the FWC is, “The demands of my family or friends interfere with work-related activities.” The internal consistencies of the WFC and FWC scales are both good (WFC Cronbach’s alpha = 0.903; M = 15.0, SD = 6.1; FWC Cronbach’s alpha = 0.866; M = 14.1, SD = 6.1).

Vacation shaming, the extent to which participants feel looked down on for taking vacation time, was assessed with items developed for this study. Each item used a 7-point Likert scale with responses ranging from strongly disagree to strongly agree. This scale included questions such as: “I feel guilty when I take vacation,” “My coworkers make comments that make me regret taking time off,” and “I experience vacation shaming at work.” Internal consistency of the vacation shaming measure is good (Cronbach’s alpha = 0.753; M = 7.9, SD = 4.5).

Absenteeism and presenteeism were assessed using single items asking participants to report the number of days (in the past 4 weeks) they missed work for health-related reasons (e.g., illness or doctor’s appointment), for dependent care, for non-health-related personal reasons, and how many days they came to work when they were sick/ill ([28]).

The Employee Turnover Questionnaire (ETQ) was used to assess overall intention to quit ([49]). Five items were drawn from this scale and were measured on a 7-point Likert scale with responses ranging from strongly disagree to strongly agree. An example of an item is, “I often think about quitting this organization.” The internal consistency of the ETQ measure is good (Cronbach’s alpha = 0.750; M = 16.1, SD = 6.3).

#### 2.2.3. Psychosocial Factors

The Perceived Stress Scale (PSS) was used to measure participants’ general stress over the past month ([14]). It includes 10 items and is based on a 5-point Likert scale with responses ranging from never to very often. An example item is: “How often have you felt difficulties were piling up so high that you could not overcome them?” The internal consistency of the PSS is good (Cronbach’s alpha = 0.797; M = 29.8, SD = 6.9).

The Interpersonal Support Evaluation List (ISEL-12) was used to measure participants’ perceived social support and included 12 items, each assessed on a 4-point Likert scale with responses ranging from definitely false to definitely true ([13]). An example item from this scale is, “If a family crisis arose, it would be difficult to find someone who could give me good advice about how to handle it.” This measure included 3 subscales: Appraisal, Belonging, and Tangible Support. The overall internal consistency of the ISEL-12 is good (Cronbach’s alpha = 0.847; M = 37.3, SD = 7.0).

Generalized Anxiety Disorder (GAD) was assessed using the Patient Health Questionnaire/Generalized Anxiety Disorder subscale (GAD-7; [69]; [42]). This measure provides a provisional diagnosis of GAD using 7 statements where participants indicate how often they experience the event, ranging from “0” (not at all) to “3” (nearly every day). The internal consistency of the GAD-7 is good (Cronbach’s alpha = 0.914; M = 10.3, SD = 5.7).

Major Depressive Disorder (MDD) was assessed using the Patient Health Questionnaire (PHQ-9; [68]), a 9-item questionnaire providing a provisional diagnosis of Major Depressive Disorder. It assessed participants’ experiences in the last 2 weeks. Questions inquire about the level of interest in doing things, feeling down or depressed, difficulty with sleeping, energy levels, eating habits, self-perception, ability to concentrate, speed of functioning, and thoughts of suicide. Responses range from “0” (Not at all) to “3” (nearly every day). The internal consistency of the PHQ-9 is good (Cronbach’s alpha = 0.905; M = 10.7, SD = 7.1).

The Maslach Burnout Inventory (MBI; [46]) is a 22-item questionnaire that measures work-related burnout and includes three subscales. The Occupational Exhaustion subscale is comprised of 9 items, such as “I feel emotionally exhausted because of my work” and “Working with people the whole day is stressful for me.” The Depersonalization/Loss of Empathy subscale comprises 5 items, such as “I have the feeling that my colleagues blame me for some of their problems” and “I’m afraid that my work makes me emotionally harder.” The Personal Accomplishment subscale includes 8 items such as “I feel full of energy” and “In my work I am very relaxed when dealing with emotional problems.” Each item is measured on a 7-point Likert scale ranging from 0 (Never) to 6 (Every day). Higher scores suggest greater frequency of the mentioned events/experiences, which indicate higher levels of exhaustion, depersonalization, and personal accomplishment, respectively. The internal consistency of each of the subscales of the MBI is good (Exhaustion Cronbach’s alpha = 0.905; M = 23.2, SD = 12.9; Personal Accomplishment Cronbach’s alpha = 0.809; M = 21.4, SD = 10.0; and Depersonalization Cronbach’s alpha = 0.702; M = 8.4, SD = 6.3).

#### 2.2.4. Analytic Approach

Total scores on the intentions to quit measure were used as the outcome variable of interest. Bivariate correlations between this measure and our occupational and psychosocial measures were calculated to determine which were related to intention to quit. Pairwise deletion was used for any missing data points. Secondly, a multiple linear regression was developed based on the significant findings at the univariate level to identify the key factors associated with intentions to quit, while controlling for mental health outcomes.[note 1] The probability of type I error was set at alpha = 0.05, and these analyses were conducted using SPSS version 29 (IBM, Inc., Chicago, IL, USA). Lastly, a path analysis was fit using the lavaan package ([60]) in R version 4.4.0 (https://www.r-project.org/), based on the Job Demands–Resources theoretical framework. This model tested whether our a priori-identified job resources and demands (organizational identification, perceived organizational support, perceived stress, and work–life balance) were associated with burnout (exhaustion subscale) and employee engagement, which then, in turn, predicted intention to quit in our sample.

## 3. Results

### 3.1. Occupational Factors

Several occupational factors were significantly related to intentions to quit (see Table 1). This included organizational identification (r = −0.33, *p* < 0.001), showing that working students in this sample who more strongly identify with their organization also tended to have a lower intention to quit. Additionally, neutral identification (r = 0.51, *p* < 0.001), ambivalent identification (r = 0.53, *p* < 0.001), and disidentification (r = 0.46, *p* < 0.001) showed positive relationships with intentions to quit, implying that the student employees who viewed their organization more negatively, or had a mixed/tense relationship with their organization tended to have increased intentions to quit. Vacation shaming (r = 0.16, *p* = 0.002) was positively related to intentions to quit, while perceived organizational support (r = −0.39, *p* < 0.001) was negatively associated with intentions to quit, suggesting those who received more shaming for taking time off work tended to have increased intention to quit, but those with more support, tended to have decreased intentions to quit. Missing work for personal reasons (but not health reasons) (r = 0.16, *p* = 0.002), as well as live–work (r = 0.36, *p* < 0.001) and work–life conflict (r = 0.31, *p* < 0.001), were all positively related to intentions to quit, suggesting that having a strong intention to quit was associated with struggling to maintain a manageable balance between work and non-work obligations. Finally, work engagement (r = −0.39, *p* < 0.001) was negatively related to intentions to quit, suggesting those who tended to be more engaged with their work also tended to have a decreased intention to quit.

### 3.2. Psychosocial Factors

Intention to quit was also significantly associated with most measured psychosocial factors (see Table 2). Perceived stress (r = 0.28, *p* < 0.001) and all subscales of the Maslach Burnout scale [Exhaustion (r = 0.40, *p* < 0.001), Accomplishment (r = 0.16, *p* < 0.001), and Depersonalization (r = 0.37, *p* < 0.001)] were positively associated with intention to quit, describing how increased stress and burnout were accompanied by increased intentions to quit. We also observed positive correlations between Generalized Anxiety Disorder (r = 0.23, *p* < 0.001) and Major Depressive Disorder (r = 0.23, *p* < 0.001) with intentions to quit; as mental health disorders increased, so did intentions to quit. Negative associations with intention to quit were observed for social support (r = −0.20, *p* < 0.001), including all subscales of this measure on the ISEL-12 [Appraisal (r = −0.13, *p* < 0.001), Belonging (r = −0.17, *p* < 0.001), and Tangible (r = −0.19, *p* < 0.001)], showing that those with higher levels of social support tended to have lower intentions to quit.

### 3.3. Multivariate Regression Analysis

Table 3 presents the results of the multiple regression analysis assessing the key factors most associated with intentions to quit in our sample. When simultaneously considering all factors associated with intention to quit at the bivariate level, this model revealed six measures that remained significantly predictive of intention to quit: neutral organizational identity (beta = 0.22, *p* < 0.001), ambivalent organizational identity (beta = 0.27, *p* < 0.001), work engagement (beta = −0.14, *p* = 0.013), life–work conflict (beta = 0.11, *p* = 0.034), perceived social support (appraisal subscale) (beta = −0.12, *p* = 0.036), and burnout (depersonalization subscale) (beta = 0.14, *p* = 0.015). The overall model was significant (F(18, 294) = 16.66, *p* < 0.001) and explained 47.5% of the variance (Adjusted R-Squared = 0.475).

### 3.4. Path Analysis

While the multiple regression analysis confirmed that burnout and work engagement were predictive of intentions to quit (while controlling for other variables and mental health outcomes like depression and generalized anxiety), this model cannot assess the more nuanced predictions of the Job Demands–Resources model. Specifically, changes in demand and resources lead to changes in employee burnout and engagement, which then predict negative outcomes like intention to quit. To determine whether the data from our sample would fit this general framework, a path model was fit with job resources (perceived organizational support and organizational identification) and demands (work–life balance and perceived stress) as covarying exogenous variables that predicted our measures of burnout (exhaustion subscale) and work engagement as mediators, which then predicted our intent to quit variable. Paths from all exogenous variables to both mediators were included based on previous findings that such cross-link paths describe working populations better than only including paths from resources to engagement and from demands to burnout ([41]).

The model fit indices for our planned model (Comparative Fit Index = 0.96, Tucker-Lewis Index = 0.93, RMSEA = 0.11, SRMR = 0.022) were acceptable. In theory, any of our exogenous variables (i.e., demands and resources) could also be directly related to Intent to Quit, and so could improve model fit. To help guide our model refinement, modification indices were estimated (using the modindices function in lavaan) with a minimum change in model fit (LRT) set to 10. This revealed one direct path that was expected to improve model fit: from Perceived Organizational Support to Intent to Quit. Freeing this path in the model improved model fit (CFI = 0.98, TLI = 0.91, RMSEA = 0.085[note 2], SRMR = 0.023), and so it was retained in the final model. No other model refinements were made (see Figure 1).

Overall, the final model fit our data reasonably well and accounted for ~29% of the variance in Intent to Quit (see Table 4). Significant direct paths were observed from Engagement (*p* < 0.001), Burnout (*p* < 0.001), and Perceived Organizational Support (*p* < 0.001) to Intent to Quit. Significant paths from Perceived Stress (*p* = 0.045), Perceived Organizational Support (*p* < 0.001), and Organizational Identification (*p* < 0.001) to Engagement were observed, accounting for 37% of the variance in Engagement. The direct path from Work–Life Conflict to Engagement was not significant (*p* = 0.39), but was retained as part of the indirect paths through Engagement and Burnout to Intent to Quit. Burnout was significantly predicted by Work–Life Conflict (*p* < 0.001) and Perceived Stress (*p* < 0.001), which accounted for 36% of the variance in Burnout. The direct paths from Organizational Identification (*p* = 0.35) and Perceived Organizational Support (*p* = 0.054) to Burnout were not significant, but were retained as part of indirect paths to Intent to Quit.

## 4. Discussion

The present study represents one of the first comprehensive explorations of occupational health among working college undergraduates, evaluating key factors associated with intentions to quit. We found increased intentions to quit were associated with poor identification with the organization, decreased work–life balance, and increased levels of burnout. Decreased intentions to quit were associated with increased work engagement and support. Notably, these findings are highly consistent with previous findings on traditional employees ([10]; [17]) and suggest working college undergraduates may well respond to organizational and psychosocial variables much like traditional employees ([39]; [76]). The results from our path analysis further suggest that the Job Demands–Resources model ([18]) explains working college undergraduates’ experiences relatively well. Similar to full-time employees, undergraduate employees’ resources (organizational identification and support) and demands (stress/strain and work–life conflict) were associated with decisions to quit.

One of the more compelling findings from our study is that increased organizational identification and support were associated with increased engagement and a decreased risk of quitting. In particular, organizational support emerged as both indirectly (through engagement) and directly related to decreased intentions to quit. This is somewhat surprising given that the working college students in our sample (from the United States) are likely to be employed for financial reasons ([55]) and contend with significant demands on their time. These results imply there may be a special role for supportive organizational practices even in the face of stress and burnout, perhaps by insulating employees from the negative feelings associated with these negative consequences. Indeed, this is consistent with recent findings that organizational support can mitigate some of the negative emotions associated with a toxic workplace by increasing both employee engagement and well-being ([57]).

Our findings also provide vital knowledge for organizations employing college students who want to implement practices to enhance employee retention. Our results suggest attrition is less likely when practices that support and engage working undergraduates are implemented. This could include practices like scheduling flexibility and supervisor support for issues that arise (e.g., [29]). Consistent with our results, we would also advocate for organizational practices that align with the beliefs and values of college student employees, as well as those that make them feel engaged and involved with their organizations. This could take many forms depending on the organization, including mission-driven practices and employee resource groups ([71]).

Unsurprisingly, decreased work–life balance was associated with increased intentions to quit in our sample, highlighting the inevitable challenge of balancing social, professional, and academic lives ([19]). This imbalance was associated with increased stress and burnout, which become major risk factors for depression and anxiety in college students ([70]). These are also factors known to be associated with increased intentions to quit in traditional employees ([43]; [61]). These psychosocial factors appear vital to occupational health but may be inherently sub-optimal in working college students. Organizations that employ college students may benefit from making their employees aware of symptoms of increased stress and burnout and the prevention and interventions that can reduce these conditions. Workplace strategies to prevent burnout can include interventions for stress management, cultivation of social support, allowing for autonomy, and providing opportunities for decision-making engagement ([23]).

Colleges and universities should also communicate the psychological vulnerabilities of working while attending college to students. Ideally, this would be done in a supportive way, where working students are made aware of available resources that can reduce the risk of burnout, such as counseling services, time management training, stress management, and relaxation training ([4]; [65]). Schools can also encourage working students to utilize social support networks to manage the stress of working while in college ([30]). These practices are particularly relevant at American institutions where almost half of full-time students and 81% of part-time students currently work during college ([55]).

Our results provide insights into the rarely studied occupational health of working college undergraduates. However, our study was limited to cross-sectional data, so we cannot make any causal claims about whether our identified predictors caused changes in intentions to quit ([18]). Follow-up longitudinal research would be required to assess such causal relationships. Longitudinal research in traditional employees has been used to support causative relationships predicted by the Job Demands–Resources model, particularly relating burnout and engagement to later intention to quit (e.g., [61]). Therefore, additional longitudinal research with working undergraduates seems especially timely, given the correlational relationships we observed in our cross-sectional data.

The current study was also limited in that it only considered undergraduate students from a traditional American four-year institution, in a sample that was gender-biased toward women. This means our findings may not generalize to other student populations, such as community college, part-time, or graduate students. Additional research is clearly needed, given that these students might prioritize work and academics differently than traditional undergraduates. For example, part-time students are more likely to work outside of school, work full-time, and have dependents ([55]). Their increased financial obligations might therefore overshadow organizational factors when considering whether or not to quit.

Given the underrepresentation of men in our sample, we also do not know if our results would generalize across genders. Though we never set out to contrast gender in our study, it is a potentially fruitful avenue for future research considering the persistent gender imbalances in certain educational tracks and professions that also tend to align with differences in coursework and time commitments ([54]). For example, men tend to outnumber women in STEM fields, which also tend to have some of the highest rates of stress and burnout ([52]). However, female students tend to report higher rates of stress, more generally (e.g., [24]), which leaves open the possibility for interesting occupational health patterns that vary by gender in working college students.

## Figures and Tables

**Figure 1 behavsci-16-00061-f001:**
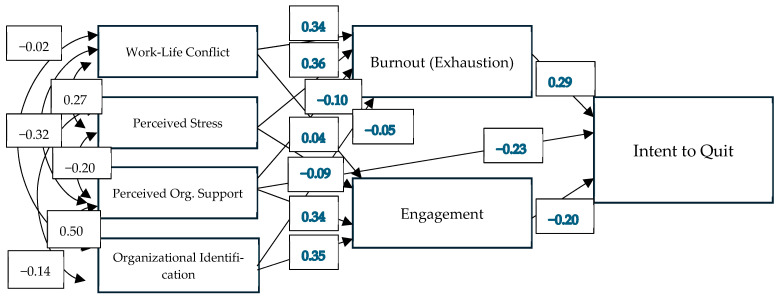
Final path analysis model with standardized coefficients (in blue) and covariances (in black).

**Table 1 behavsci-16-00061-t001:** Descriptive summary statistics of occupational factors and correlations with intention to quit.

Variable	Min	Max	Mean (SD)	Pearson Correlation Coefficient	*p*-Value
Intention to Quit	5	35	16.12 (6.26)	--	--
Organizational Identity					
Org ID Composite	4	20	13.56 (3.19)	−0.328	*p* < 0.001 *
Neutral Org ID	6	30	14.95 (6.14)	0.505	*p* < 0.001 *
Disidentification Org ID	6	30	9.29 (4.68)	0.458	*p* < 0.001 *
Ambivalent Org ID	6	30	12.18 (5.91)	0.526	*p* < 0.001 *
Vacation Shaming	3	21	7.85 (4.49)	0.155	*p* = 0.002 *
Perceived Organizational Support	8	40	28.83 (7.2)	−0.391	*p* < 0.001 *
Absenteeism/Presenteeism					
Absent—Health Reasons	0	7	0.35 (0.84)	0.021	*p* = 0.685
Absent—Personal Reasons	0	5	0.58 (1.03)	0.16	*p* = 0.002 *
Present at work while sick	0	20	1.84 (3.38)	0.052	*p* = 0.315
Work Engagement	17	63	48.09 (8.60)	−0.388	*p* < 0.001 *
Work–Life Conflict	4	28	15.02 (6.15)	0.305	*p* < 0.001 *
Live–Work Conflict	5	35	14.07 (6.11)	0.358	*p* < 0.001 *

* indicates significant effect with *p* < 0.05.

**Table 2 behavsci-16-00061-t002:** Descriptive summary statistics of psychosocial factors and correlations with intention to quit.

Variable	Min	Max	Mean (SD)	Pearson Correlation Coefficient	*p*-Value
Intention to Quit	5	35	16.12 (6.26)	--	--
Perceived Stress Scale	10	50	29.85 (6.89)	0.278	*p* < 0.001 *
Generalized Anxiety Disorder Total Score	0	21	10.32 (5.74)	0.232	*p* < 0.001 *
Major Depressive Disorder Total Score	0	27	10.71 (7.10)	0.225	*p* < 0.001 *
Social Support (ISEL-12)					
Total Score	16	48	37.30 (7.03)	−0.197	*p* < 0.001 *
Appraisal Subscale	4	16	12.68 (2.86)	−0.127	*p* < 0.001 *
Belonging Subscale	4	16	12.16 (2.92)	−0.173	*p* < 0.001 *
Tangible Subscale	5	16	12.46 (2.43)	−0.189	*p* < 0.001 *
Maslach Burnout Score					
Exhaustion Subscale	0	54	23.21 (12.88)	0.401	*p* < 0.001 *
Accomplishment Subscale	0	48	21.44 (9.97)	0.164	*p* < 0.001 *
Depersonalization Subscale	0	30	8.36 (6.32)	0.369	*p* < 0.001 *
Self-Esteem	9	20	13.95 (1.77)	0	*p* = 0.997

* indicates significant effect with *p* < 0.05.

**Table 3 behavsci-16-00061-t003:** Multivariate linear regression assessing occupational and psychosocial factors associated with intentions to quit (* indicates *p* < 0.05).

Variable	*b*	SE	Beta	*p*-Value	95% Lower CI	95% Upper CI
(Constant)	4.576	3.772		0.226	−2.848	12.000
Organizational Identification Total	−0.110	0.104	−0.057	0.294	−0.315	0.096
Org ID = Neutral	0.223	0.055	0.218	<0.001 *	0.115	0.331
OrgID = Disidentification	0.103	0.077	0.078	0.182	−0.049	0.256
OrgID = Ambivalent	0.287	0.066	0.273	<0.001 *	0.158	0.417
Vacation Shaming Total	0.035	0.062	0.025	0.577	−0.087	0.156
Perceived Organization Support Total	0.056	0.054	0.064	0.301	−0.050	0.162
Work Engagement Total	−0.102	0.041	−0.140	0.013 *	−0.182	−0.022
Work–Life Conflict Total	−0.002	0.057	−0.002	0.966	−0.116	0.111
Live–Work Conflict Total	0.117	0.055	0.114	0.034 *	0.009	0.225
Perceived Stress Scale Total	0.062	0.056	0.069	0.268	−0.048	0.173
Gen Anxiety Disorder—Total Score	0.018	0.070	0.017	0.798	−0.120	0.156
Major Depressive Disorder—Total	−0.052	0.057	−0.058	0.359	−0.165	0.060
Perceived Social Support—Appraisal	0.269	0.128	−0.121	0.036 *	0.017	0.522
Perceived Social Support—Belonging	−0.157	0.121	−0.074	0.194	−0.395	0.080
Perceived Social Support—Tangible	0.047	0.161	0.018	0.768	−0.269	0.364
Burnout—Exhaustion	0.034	0.034	0.071	0.310	−0.032	0.100
Burnout—Personal Accomplishment	0.046	0.034	0.073	0.174	−0.020	0.112
Burnout-Depersonalization	0.145	0.059	0.143	0.015 *	0.029	0.261

Model Summary R = 0.711, R^2^ = 0.505, Adjusted R^2^ = 0.475, F(18, 294) = 16.664, *p* < 0.001.

**Table 4 behavsci-16-00061-t004:** Results of Job Demands–Resources path model.

Outcome Variable Predictor	Estimate	S.E.	Z-Value	*p*-Value	Standardized Coefficient
Engagement (R^2^ = 0.37)					
Perceived Org. Support	0.41	0.063	6.54	<0.001 *	0.34
Org. Identification	0.93	0.130	6.96	<0.001 *	0.35
Perceived Stress	−0.11	0.056	−2.00	0.045 *	−0.09
Work–Life Conflict	0.06	0.066	0.86	0.39	0.04
Burnout (Exhaustion) (R^2^ = 0.36)					
Perceived Org. Support	−0.18	0.095	−1.93	0.054	−0.10
Org. Identification	−0.19	0.200	−0.93	0.35	−0.05
Perceived Stress	0.67	0.085	7.92	<0.001 *	0.36
Work–Life Conflict	0.71	0.099	7.20	<0.001 *	0.34
Intention to Quit (R^2^ = 0.29)					
Engagement	−0.16	0.038	−4.33	<0.001 *	−0.23
Burnout	0.14	0.023	6.18	<0.001 *	0.29
Perceived Org. Support	−0.17	0.047	−3.64	<0.001 *	−0.20
Covariances					
Perceived Org. Support and Org. Identification	11.52	1.39	8.32	<0.001 *	0.50
Org. Identification and Work–Life Conflict	−0.32	1.06	−0.30	0.76	−0.02
Org. Identification and Perceived Stress	−3.19	1.20	−2.66	0.008 *	−0.14
Perceived Stress and Work–Life Conflict	11.33	2.33	4.86	<0.001 *	0.27
Perceived Org. Support and Work–Life Conflict	−14.02	2.45	−5.68	<0.001 *	−0.32
Perceived Org. Support and Perceived Stress	−10.01	2.69	−3.72	<0.001 *	−0.20

Model fit indices: CFI = 0.98, TLI = 0.91, RMSEA = 0.085, SRMR = 0.023. * indicated *p* < 0.05.

## Data Availability

The data obtained for this study are publicly available on the Texas Data Repository: https://doi.org/10.18738/T8/MWISWN.

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
