# Peer review of "Between Class and Career: Applying the Job Demands–Resources Model to Working College Students"

_behavsci, 2025, doi:10.3390/bs16010061_

Round 1
Reviewer 1 Report
Comments and Suggestions for Authors
The study addresses an important and highly relevant topic: the job engagement, burnout, and turnover intentions of working university students within the framework of the Job Demands-Resources (JD-R) model. Investigating this population in such depth is particularly significant, as working students remain an underexplored group in the existing literature. The manuscript is logically organized, well structured, and the statistical procedures are appropriately applied. The conclusions are consistent with the empirical findings, and the practical implications are clearly articulated. Nonetheless, several areas could benefit from further refinement.
It would be advisable to briefly present the characteristics of the sample in the abstract, as this information is essential for interpreting the study. Although the article draws primarily on psychological and organizational psychology sources, it would be beneficial to incorporate insights from additional disciplines -such as higher education studies, labour market sociology, and dropout research-which are also highly relevant to the topic. Some of the cited studies on student employment are considerably outdated, often close to a decade old, despite the fact that numerous recent publications have addressed both student employment and its relationship with academic dropout.
It is also important to highlight that the prevalence and motivations of student employment, as well as its impact on academic progress, vary substantially depending on the structure of a country’s higher education system (e.g. tuition-free access, financial support schemes, national dropout prevention strategies) and the field of study. International datasets- such as the Eurostudent surveys-show that in certain disciplines, particularly in health sciences, the proportion of working students is considerably lower. The current theoretical background emphasizes mainly the negative consequences of student employment, while the broader literature describes its dual nature: working during studies can have both positive and negative effects, often conceptualized as a “double-edged sword.”
The methodological section currently lacks clearly formulated research questions and hypotheses. Although the manuscript references theoretical expectations based on the JD–R model, these are not explicitly stated. Presenting 3–5 precisely defined hypotheses at the end of the introduction would improve the clarity and coherence of the study’s analytic framework.
It would also be helpful to describe the conditions of data collection in greater detail, including the time (year, semester), mode, and context of the data gathering. In addition, more information concerning participants in the broader “occupational health study” would allow the reader to better situate the present sample. It would be relevant to know whether the working students were enrolled in bachelor’s or master’s programmes, as this influences the likelihood of employment, the type of work undertaken, and students’ motivations. If available, incorporating information on the reasons for working (e.g. financial necessity versus gaining professional experience) would further strengthen the analysis, since working out of financial necessity is a well-documented risk factor.
A deeper discussion of potential biases due to sample homogeneity would also be warranted. The sample consists of 80% women and includes only first-year psychology students, which substantially limits the generalizability of the findings. It would be useful to elaborate on how gender distribution and disciplinary background might influence stress, burnout, or engagement indicators.
The presentation of measurement tools and scales is appropriate, detailed, and methodologically sound. However, the regression model includes 18 predictors, which may overburden the model even with multicollinearity checks in place. A more detailed reflection on the researchers’ modelling decisions and potential sources of bias would be valuable, as well as consideration of whether simplified models or alternative analytical strategies (e.g. factor structures, variable reduction) might be justified.
Regarding limitations, the authors acknowledge the constraints associated with the cross-sectional design, but this point should be emphasized more strongly, particularly because the causal interpretations implied in the path analysis may be misleading in the absence of longitudinal data.
Author Response
Thanks to both of the reviewers for their time and effort in reviewing our manuscript, and for their highly considered and constructive feedback. We found it to be fair, thorough, and incredibly helpful in improving the overall quality of our manuscript.
We did our best to balance calls across the two reviewers for editing down text to be more concise while also offering more elaborate explanations on certain important points. We feel the guidance we have received from the reviewers has enabled us to greatly improve the presentation of our study.
Reviewer 1’s comments (and our responses):
- It would be advisable to briefly present the characteristics of the sample in the abstract, as this information is essential for interpreting the study.
- Response/Action Taken: Thank you for this suggestion. We have added demographic information about our sample to the abstract: “The current study assessed organizational and psychosocial factors related to intentions to quit in American working college undergraduates (N = 382; mean age = 19 years; ~80% female), an understudied population in occupational health research.” (p. 1, lines 13-14).”
- Although the article draws primarily on psychological and organizational psychology sources, it would be beneficial to incorporate insights from additional disciplines -such as higher education studies, labour market sociology, and dropout research-which are also highly relevant to the topic.
- Response/Action Taken: While editing the Introduction and Discussion we endeavored to included sources from various disciplines (e.g., higher education, labor market studies, etc.,), as you suggested. These new references are peppered throughout the paper, and hopefully add to the breadth in relevant topics.
- Some of the cited studies on student employment are considerably outdated, often close to a decade old, despite the fact that numerous recent publications have addressed both student employment and its relationship with academic dropout.
- Response/Action Taken: We have now updated many of the older references cited in the Introduction, as well as in the Discussion, so that our claims about student experiences and outcomes rely on recent findings from the literature. These are included in the text and the References section in blue font, so that these changes are easy to spot.
- It is also important to highlight that the prevalence and motivations of student employment, as well as its impact on academic progress, vary substantially depending on the structure of a country’s higher education system (e.g. tuition-free access, financial support schemes, national dropout prevention strategies) and the field of study. International datasets- such as the Eurostudent surveys-show that in certain disciplines, particularly in health sciences, the proportion of working students is considerably lower.
- Response/Action Taken: Thank you for this nuanced perspective. We have now included text in the Introduction to highlight this distinction and its importance: “This nuance is important to consider given most students in countries that do not offer national tuition assistance work more than part time (Perna, 2023), whereas students in many European countries devote fewer than 20 hours per week working outside school (Hauschildt et al., 2024). The current study focuses on a sample of college undergraduates from the United States, where the vast majority work for financial reasons (Perna).” (p. 2, lines 53-58)
- The current theoretical background emphasizes mainly the negative consequences of student employment, while the broader literature describes its dual nature: working during studies can have both positive and negative effects, often conceptualized as a “double-edged sword.”
- Response/Action Taken: Thank you for this perspective. It is important to consider the benefits of college student employment. We have now added a paragraph to the Introduction to make this point: “Employment during college is appealing because it brings advantages like helping students pay for tuition and related expenses. Graduates who work during college also tend to earn higher salaries, even after controlling for other variables (Perna, 2023). As Remenick and Bergman (2021) explain, work can help students apply what they learn in class as well as develop soft skills and networks that are useful in future professional endeavors. Several studies have also demonstrated that student employment helps with persistence towards degree completion (Kulm & Cramer, 2006; Martinez et al., 2012). Research on the effect of employment on GPA and academic performance is mixed, but evidence suggests part-time employment and working on campus yield greater benefits than full-time work off-campus (Dundes & Marx, 2006; McClellan et al., 2023). This nuance is important to consider given most students in countries that do not offer national tuition assistance work more than part time (Perna, 2023), whereas students in many European countries devote fewer than 20 hours per week working outside school (Hauschildt et al., 2024). The current study focuses on a sample of college undergraduates from the United States, where the vast majority work for financial reasons (Perna).” (p. 2, lines 44-58).
- The methodological section currently lacks clearly formulated research questions and hypotheses. Although the manuscript references theoretical expectations based on the JD–R model, these are not explicitly stated. Presenting 3–5 precisely defined hypotheses at the end of the introduction would improve the clarity and coherence of the study’s analytic framework.
- Response/Action Taken: We apologize if our research questions were not as concrete as they could be. We have added specific predictions to the end of the Introduction section (as you suggested) to clarify what we are testing: “Specifically, we use multiple regression and path analysis to test the following predictions about our working undergraduate sample:
- Increased levels of stress, work-life conflict, absenteeism, and burnout will be associated with increased intentions to quit.
- Increased social support, self-esteem, engagement, and organizational identification will be associated with decreased intentions to quit.
- Demands (e.g., stress and work-life conflict) will be positively associated with burnout, which in turn, will be positively associated with intention to quit.
- Resources (e.g., organizational support and identification) will be positively associated with engagement, which in turn, will be negatively associated with intention to quit.” (p. 4, lines 161-171)
- Response/Action Taken: We apologize if our research questions were not as concrete as they could be. We have added specific predictions to the end of the Introduction section (as you suggested) to clarify what we are testing: “Specifically, we use multiple regression and path analysis to test the following predictions about our working undergraduate sample:
- It would also be helpful to describe the conditions of data collection in greater detail, including the time (year, semester), mode, and context of the data gathering.
- Response/Action Taken: We apologize for the lack of procedural information provided in the original manuscript. We have now edited the Participants and Procedures subsection to include the missing information you identified: “Students enrolled in Introductory Psychology during the 2023 Fall semester, were required to engage in research experience options, one of which was to participate in a research study. Students who met the study’s criteria (being employed while being enrolled in college courses) and chose to participate, were asked to complete an online, anonymous survey in Qualtrics.” (p. 4, lines 183-187)
- In addition, more information concerning participants in the broader “occupational health study” would allow the reader to better situate the present sample.
- Response/Action Taken: We have now added more information about the larger study that included working adults of all types (not just students) in the Participants and Procedures subsection: “Participants (N = 382) were comprised of a sub-sample of self-identified working college undergraduates from a larger study on the occupational health of working adults that focused on psychological and occupational factors related to engagement/disengagement, intention to quit, and perceptions of quiet quitting.” (p. 4, lines 174-178)
- It would be relevant to know whether the working students were enrolled in bachelor’s or master’s programmes, as this influences the likelihood of employment, the type of work undertaken, and students’ motivations.
- Response/Action Taken: This is a great point. We do mention briefly in the Discussion that our findings may differ in other types of students, like graduate students, but it certainly makes a lot of sense to clearly identify our participants as undergraduates from the start. This has now been added to the abstract, the Participants and Procedures subsection, mentioned above, and other places throughout the paper.
- If available, incorporating information on the reasons for working (e.g. financial necessity versus gaining professional experience) would further strengthen the analysis, since working out of financial necessity is a well-documented risk factor.
- Response/Action Taken: This is an important point. While we did not directly assess the reason for working in our survey, we have internal institutional data indicating that the majority of students at our institution work for financial reasons (~70%). We have now included information to clarify that participants in our sample primarily work for financial reasons (see p. 2, lines 57-58).
- A deeper discussion of potential biases due to sample homogeneity would also be warranted. The sample consists of 80% women and includes only first-year psychology students, which substantially limits the generalizability of the findings. It would be useful to elaborate on how gender distribution and disciplinary background might influence stress, burnout, or engagement indicators.
- Response/Action Taken: This is an important limitation that was also noted by Reviewer 2. We have now added additional text to the Discussion that elaborates on this limitation and the subsequent lack of generalizability of our results: “The current study was also limited in that it only considered undergraduate students from a traditional American four-year institution, in a sample that was gender-biased toward women. This means our findings may not generalize to other student populations such as community college, part-time, or graduate students. Additional research is clearly needed given that these students might prioritize work and academics differently than traditional undergraduates. For example, part-time students are more likely to work outside of school, work full time, and have dependents (Perna, 2023). Their increased financial obligations might therefore overshadow organizational factors when considering whether or not to quit.
Given the underrepresentation of men in our sample, we also do not know if our results would generalize across genders. Though we never set out to contrast gender in our study, it is a potentially fruitful avenue for future research considering the persistent gender imbalances in certain educational tracks and professions that also tend to align with differences in coursework and time commitments (Patnaik et al., 2021). For example, men tend to outnumber women in STEM fields, which also tend to have some of the highest rates of stress and burnout (Olson et al., 2025). However, female students tend to report higher rates of stress, more generally (e.g., Graves et al., 2021), which leaves open the possibility for interesting occupational health patterns that vary by gender in working college students.” (p. 14, lines 498-516)
- The presentation of measurement tools and scales is appropriate, detailed, and methodologically sound. However, the regression model includes 18 predictors, which may overburden the model even with multicollinearity checks in place. A more detailed reflection on the researchers’ modelling decisions and potential sources of bias would be valuable, as well as consideration of whether simplified models or alternative analytical strategies (e.g. factor structures, variable reduction) might be justified.
- Response/Action Taken: While we appreciate your point here, we want to highlight the fact that this study was partially exploratory, and we wanted to be as agnostic as possible as to the which factors emerged as significantly related to our outcome variable of interest. We also were prioritizing breadth when considering both occupational and psychosocial factors, as a priori, we viewed them as equally important to consider. We therefore have opted to retain the final (simultaneous) multiple regression model (given the fit was good and we experienced no issues with multicollinearity). However, given what we have learned in this initial study, we do think it would be worthwhile to conduct follow-up research that can be analytically more targeted in inclusion/exclusion of predictor variables.
- Regarding limitations, the authors acknowledge the constraints associated with the cross-sectional design, but this point should be emphasized more strongly, particularly because the causal interpretations implied in the path analysis may be misleading in the absence of longitudinal data.
- Response/Action Taken: We appreciate your point, here, and certainly do not want to overextend the interpretation of our results in a way that isn’t warranted given our design. We have moved up and revised the paragraph in the Discussion addressing this limitation to make this point more prominent and less ambiguous: “Our results provide insights into the rarely studied occupational health of working college undergraduates. However, our study was limited to cross-sectional data, so we cannot make any causal claims about whether our identified predictors caused changes in intentions to quit (Demoerouti et al., 2001). Follow-up longitudinal research would be required to assess such causal relationships. Longitudinal research in traditional employees has been used to support causative relationships predicted by the Job Demands-Resources model, particularly relating burnout and engagement to later intention to quit (e.g., Rouleau et al., 2012). Therefore, additional longitudinal research with working undergraduates seems especially timely, given the correlational relationships we observed in our cross-sectional data.” (pp. 14, lines 488-497).
Reviewer 2 Report
Comments and Suggestions for Authors
Thank you for the opportunity to review this manuscript. The paper addresses an important and understudied topic—the occupational health and turnover intentions of working college students—by applying the Job Demands–Resources (JD-R) model to this population. Overall, the manuscript is well-written, theoretically grounded, and supported by a comprehensive set of measures. The study has potential to make a meaningful contribution to the literature. Below, I outline strengths as well as areas that would benefit from clarification or revision.
The manuscript addresses a clear gap in the literature, as working college students are rarely examined through the JD-R framework.
The theoretical background is rich and engages well with both classic and contemporary scholarship.
The methodology is clearly described, with appropriate measures and acceptable reliability indices.
The statistical analyses (correlations, regression, and path analysis) are rigorous and well-presented.
Tables and figures are clear, well-formatted, and aid interpretation.
Areas for improvement:
- The introduction offers valuable context but is overly long and could be more succinct. Several paragraphs repeat concepts or contain detail that could be moved to the Discussion. Streamlining this section would improve clarity and focus.
- While the study is exploratory, the manuscript would benefit from explicit research questions or hypotheses derived from the JD-R model. This would provide a clearer framework for understanding the analytical choices and interpreting the results.
- Some interpretations come close to implying causality, even though the data are cross-sectional. These passages should be revised to emphasize associations rather than effects. Additionally, the Discussion could more directly explain why certain predictors (e.g., organizational support) maintain direct effects while others do not.
- The path model shows acceptable fit, but the RMSEA is moderately high. It would strengthen the manuscript to acknowledge this limitation and consider discussing alternative model specifications or theoretical explanations for the weaker fit.
- The Discussion is coherent but could be more concise. Some implications (particularly those concerning interventions) could be better tied to the specific patterns observed in the data. It would also be beneficial to reflect more deeply on the conceptual meaning of engagement emerging as a strong predictor compared to burnout.
- Although limitations are acknowledged, the restricted sampling frame (Introductory Psychology students at one institution) deserves more emphasis, particularly regarding generalizability.
Some paragraphs contain long sentences that could be simplified for readability.
Consider reorganizing the flow of the Measures section to avoid redundancy.
This manuscript provides important insights into the occupational health experiences of working college students and represents a valuable application of the JD-R model. With revisions to improve clarity, theoretical alignment, and interpretative caution, the manuscript will be significantly strengthened and more impactful.
Author Response
Thanks to both of the reviewers for their time and effort in reviewing our manuscript, and for their highly considered and constructive feedback. We found it to be fair, thorough, and incredibly helpful in improving the overall quality of our manuscript.
We did our best to balance calls across the two reviewers for editing down text to be more concise while also offering more elaborate explanations on certain important points. We feel the guidance we have received from the reviewers has enabled us to greatly improve the presentation of our study.
Reviewer 2’s comments (and our responses):
- The introduction offers valuable context but is overly long and could be more succinct. Several paragraphs repeat concepts or contain detail that could be moved to the Discussion. Streamlining this section would improve clarity and focus.
- Response/Action Taken: We have edited and restructured the Introduction to be more concise and hopefully flow better. Additionally, we have added to the Discussion in the ways suggested by you and the other reviewer. We believe both of these sections are now clearer and more cohesive.
- While the study is exploratory, the manuscript would benefit from explicit research questions or hypotheses derived from the JD-R model. This would provide a clearer framework for understanding the analytical choices and interpreting the results.
- Response/Action Taken: Thank you for this suggestion that was consistent with that of Reviewer 1. As mentioned in response to that reviewer, we have now added specific predictions to the end of the Introduction section (as you suggested) to clarify what we are testing: “Specifically, we use multiple regression and path analysis to test the following predictions about our working undergraduate sample:
- Increased levels of stress, work-life conflict, absenteeism, and burnout will be associated with increased intentions to quit.
- Increased social support, self-esteem, engagement, and organizational identification will be associated with decreased intentions to quit.
- Demands (e.g., stress and work-life conflict) will be positively associated with burnout, which in turn, will be positively associated with intention to quit.
- Resources (e.g., organizational support and identification) will be positively associated with engagement, which in turn, will be negatively associated with intention to quit.” (p. 4, lines 161-171)
- Response/Action Taken: Thank you for this suggestion that was consistent with that of Reviewer 1. As mentioned in response to that reviewer, we have now added specific predictions to the end of the Introduction section (as you suggested) to clarify what we are testing: “Specifically, we use multiple regression and path analysis to test the following predictions about our working undergraduate sample:
- Some interpretations come close to implying causality, even though the data are cross-sectional. These passages should be revised to emphasize associations rather than effects.
- Response/Action Taken: We have edited the wording in the Results and Discussion sections to ensure we are never claiming causation, as our results only show associations. As mentioned to our response to reviewer 1, we have also made the paragraph explaining the cross-sectional nature of our data (and the limitations of this) more prominent in the Discussion section.
- Additionally, the Discussion could more directly explain why certain predictors (e.g., organizational support) maintain direct effects while others do not.
- Response/Action Taken: Thank you for this suggestion. In revising the Discussion section, we have enhanced our discussion of organizational support as a prominent predictive variable: “One of the more compelling findings from our study is that increased organizational identification and support were associated with increased engagement and a decreased risk of quitting. In particular, organizational support emerged as both indirectly (through engagement) and directly related to decreased intentions to quit. This is somewhat surprising given the working college students in our sample (from the United States) are likely to be employed for financial reasons (Perna, 2023), and contend with significant demands on their time. These results imply there may be a special role for supportive organizational practices even in the face of stress and burnout, perhaps by insulating employees from the negative feelings associated with these negative consequences. Indeed, this is consistent with recent findings that the organizational support can mitigate some of the negative emotions associated with a toxic workplace by increasing both employee engagement and well-being (Rasool et al., 2021).” (see p.14, lines 444-455)
- The path model shows acceptable fit, but the RMSEA is moderately high. It would strengthen the manuscript to acknowledge this limitation and consider discussing alternative model specifications or theoretical explanations for the weaker fit.
- Response/Action Taken: This is a fair criticism. When estimating the path model, we were trying to be conservative in fitting the model based primarily on the theoretical framework of the Job Demands-Resources model and the path analyses from previous research using this model. It is true that during model fitting, there were cross-link paths, particularly from the Resources variables to the Burnout variable that were not a great fit for our data. Anecdotally, removing these does decrease the RMSEA. However, we were trying to avoid tinkering with the model based solely on our data, and as they were theoretically motivated and not flagged in the modification indices we calculated, we opted to keep these paths. We have added a footnote explaining this decision: “We acknowledge the RMSEA is slightly higher than 0.08 cut-off, indicating a good fit. This is likely due to including paths from our Resource variables (organizational support and organizational identification) to Burnout. These non-significant paths were retained to adhere to the a priori theoretical model we intended to test, though removal of them does improve the RMSEA of the overall model.” ( 11).
- The Discussion is coherent but could be more concise. Some implications (particularly those concerning interventions) could be better tied to the specific patterns observed in the data. It would also be beneficial to reflect more deeply on the conceptual meaning of engagement emerging as a strong predictor compared to burnout.
- Response/Action Taken: We have now extensively edited the Discussion section to tie our recommended interventions more specifically to our obtained results as well as to discuss engagement and burnout separately. However, we note that in both the multiple regression and the path model, engagement and burnout were roughly equally predictive of/associated with intentions to quit. Thus, we have not contrasted engagement to burnout as a stronger predictor as you recommended. We just don’t feel our data warrant this.
- Although limitations are acknowledged, the restricted sampling frame (Introductory Psychology students at one institution) deserves more emphasis, particularly regarding generalizability.
- Response/Action Taken: We acknowledge this oversight (that was also brought up by Reviewer 1), and have added additional text to the Discussion that consider the consequences of this limitation: “The current study was also limited in that it only considered undergraduate students from a traditional American four-year institution, in a sample that was gender-biased toward women. This means our findings may not generalize to other student populations such as community college, part-time, or graduate students. Additional research is clearly needed given that these students might prioritize work and academics differently than traditional undergraduates. For example, part-time students are more likely to work outside of school, work full time, and have dependents (Perna, 2023). Their increased financial obligations might therefore overshadow organizational factors when considering whether or not to quit.
Given the underrepresentation of men in our sample, we also do not know if our results would generalize across genders. Though we never set out to contrast gender in our study, it is a potentially fruitful avenue for future research considering the persistent gender imbalances in certain educational tracks and professions that also tend to align with differences in coursework and time commitments (Patnaik et al., 2021). For example, men tend to outnumber women in STEM fields, which also tend to have some of the highest rates of stress and burnout (Olson et al., 2025). However, female students tend to report higher rates of stress, more generally (e.g., Graves et al., 2021), which leaves open the possibility for interesting occupational health patterns that vary by gender in working college students.” (p. 14, lines 498-516)
- Some paragraphs contain long sentences that could be simplified for readability.
- Response/Action Taken: As part of your earlier comment about making the Introduction and Discussion more concise, we have edited the entire document to improve the clarity and conciseness of our writing.
- Consider reorganizing the flow of the Measures section to avoid redundancy.
- Response/Action Taken: We have critically read through the information provided in the Measures subsection to ensure there is no redundancy. We include only one explanation for each unique measure included in our analyses. We have decided to retain the current presentation of measures to preserve the grouping based on occupational versus psychosocial variables.
Round 2
Reviewer 1 Report
Comments and Suggestions for Authors
The authors have responded thoroughly and constructively to the reviewer’s comments. The revised manuscript now includes a clearer presentation of the sample characteristics in the abstract, explicitly formulated research questions and hypotheses grounded in the JD–R framework, and a more detailed description of the data collection context. The theoretical background has been strengthened with more recent and relevant literature on student employment, and the limitations related to sample homogeneity and the cross-sectional design are now clearly and appropriately acknowledged.
While the manuscript remains primarily rooted in a psychological perspective and offers limited reflection on the complexity of the multivariate modelling strategy, these issues do not substantially detract from the overall quality of the revision. Overall, the authors have addressed the main concerns raised in the review, resulting in a clearer, more transparent, and methodologically stronger manuscript.
Reviewer 2 Report
Comments and Suggestions for Authors
Dear authors,
Thank you for submitting the revised version of the manuscript and for the detailed responses to the reviewers’ comments. After an integrated reading of both the revised article and the changes made, I consider that the manuscript has been adequately revised.
The main issues raised by the reviewers have been appropriately addressed, namely:
(i) improved clarity and focus of the Introduction, including the explicit formulation of predictions grounded in the JD-R model;
(ii) careful use of associative language consistent with the cross-sectional design of the study;
(iii) a more developed and better-integrated Discussion, closely aligned with the empirical results; and
(iv) explicit acknowledgement of the limitations of both the model and the sample, particularly regarding generalisability.
Overall, the manuscript now demonstrates stronger theoretical coherence, analytical clarity, and interpretative maturity, and it is aligned with the expected scientific standards. From my perspective, the manuscript is suitable to proceed in the editorial process.